# Real-World Clinical Outcomes Associated with Canagliflozin in Patients with Type 2 Diabetes Mellitus in Spain: The Real-Wecan Study

**DOI:** 10.3390/jcm9072275

**Published:** 2020-07-17

**Authors:** Juan J. Gorgojo-Martínez, Manuel A. Gargallo-Fernández, Alba Galdón Sanz-Pastor, Teresa Antón-Bravo, Miguel Brito-Sanfiel, Jaime Wong-Cruz

**Affiliations:** 1Department of Endocrinology and Nutrition, Hospital Universitario Fundación Alcorcón, Alcorcón, 28922 Madrid, Spain; jemwong@fhalcorcon.es; 2Department of Endocrinology and Nutrition, Hospital Universitario Infanta Leonor, 28031 Madrid, Spain; gargallomgar@gmail.com; 3Department of Endocrinology and Nutrition, Fundación Jiménez Díaz, 28040 Madrid, Spain; albagaldonsp@gmail.com; 4Department of Endocrinology and Nutrition, Hospital Universitario de Móstoles, Móstoles, 28935 Madrid, Spain; tantonb1978@gmail.com; 5Department of Endocrinology and Nutrition, Hospital Universitario Puerta de Hierro, Majadahonda, 28222 Madrid, Spain; mbritosanfiel@gmail.com

**Keywords:** SGLT-2 inhibitor, canagliflozin, type 2 diabetes, switch

## Abstract

The aims of this multicentric retrospective study were to assess in a real-world setting the effectiveness and safety of canagliflozin 100 mg/d (CANA100) as an add-on to the background antihyperglycemic therapy, and to evaluate the intensification of prior sodium–glucose co-transporter type 2 inhibitor (SGLT-2i) therapy by switching to canagliflozin 300 mg/d (CANA300) in patients with T2DM. One cohort of SGLT2i-naïve patients with T2DM who were initiated on CANA100 and a second cohort of patients with prior background SGLT-2i therapy who switched to CANA300 were included in the study. The primary outcome of the study was the mean change in HbA1c over the follow-up time. In total, 583 patients were included—279 in the cohort of CANA100 (HbA1c 8.05%, weight 94.9 kg) and 304 in the cohort of CANA300 (HbA1c 7.51%, weight 92.0 kg). Median follow-up periods in both cohorts were 9.1 and 15.4 months respectively. CANA100 was associated to significant reductions in HbA1c (−0.90%) and weight (−4.1 kg) at the end of the follow-up. In those patients with baseline HbA1c > 8% (mean 9.25%), CANA100 lowered HbA1c levels by 1.51%. In the second cohort, patients switching to CANA300 experienced a significant decrease in HbA1c (−0.35%) and weight (−2.1 kg). In those patients with baseline HbA1c > 8% (mean 8.94%), CANA300 lowered HbA1c levels by 1.12%. There were significant improvements in blood pressure in both cohorts. No unexpected adverse events were reported. In summary, CANA100 (as an add-on therapy) and CANA300 (switching from prior SGLT-2i therapy) significantly improved several cardiometabolic parameters in patients with T2DM.

## 1. Introduction

Sodium–glucose co-transporter type 2 inhibitors (SGLT-2is) are a class of antihyperglycemic drugs that reduce tubular reabsorption of filtered glucose, inducing urinary glucose excretion (UGE) [1]. Glucosuria causes a decrease in fasting and postprandial plasma glucose and promotes weight and fat mass reductions secondary to calorie loss through urine. Blocking tubular glucose reabsorption facilitates its exchange by uric acid through human glucose transporter 9 (GLUT-9), inducing uricosuria and reduction of uricemia. In addition, sodium–glucose co-transporter type 2 SGLT-2 inhibition leads to natriuresis and mild osmotic diuresis, which contributes to a significant decrease in systolic and diastolic blood pressure (BP) [2].

Four SGLT-2is (dapagliflozin, empagliflozin, canagliflozin, and ertugliflozin) have been marketed in the European Union so far. In patients with T2DM, these drugs induce a UGE ranging between 60 to 80 g per day, although greater levels of glucosuria (above 110 g/day) have been observed with high doses of canagliflozin [3]. Unlike other SGLT2-is, which show a high specificity for SGLT-2, canagliflozin 300 mg (CANA300) induces a transient inhibition of the sodium–glucose co-transporter type 1 (SGLT-1) in the intestine, which reduces postprandial blood glucose and stimulates distal secretion of glucagon-like peptide-1 (GLP-1) and peptide tyrosine tyrosine (PYY), probably as a result of intraluminal glucose metabolism by the gut microbiome to short-chain fatty acids, which subsequently stimulates L cell secretion [4,5,6].

In randomized clinical trials (RCTs) conducted in patients with type 2 diabetes (T2DM), SGLT-2is have shown superiority or at least non-inferiority in glycemic control compared to other oral glucose-lowering drugs (GLDs), achieving greater weight loss (WL) and BP reduction [7,8,9,10,11,12,13,14]. The incidence of hypoglycemia with SGLT-2is was similar to that observed with placebo. In addition to these glycemic and metabolic improvements, SGLT-2is have demonstrated cardiovascular (CV) and renal benefits in patients with T2DM and high cardiovascular (CV) risk or chronic kidney disease (CKD) [15,16,17,18,19].

RCTs with canagliflozin have shown a dose–response relationship with glycated hemoglobin (HbA1c) and WL, with doses ranging from 50 to 300 mg per day [20]. However, to date, no head-to-head RCTs comparing different SGLT-2is in patients with T2DM have been reported. A network meta-analysis performing indirect comparisons from phase 3 RCTs of each drug showed that CANA300 induced greater reductions in HbA1c, weight, and systolic BP than dapagliflozin or empagliflozin, although these differences were modest [21]. A higher effect on UGE and postprandial glycemia could explain the better results seen with CANA300.

While RCTs remain the gold standard for drug approval, real-world studies (RWS), although statistically less rigorous, can provide valuable insight into how GLDs perform within specific subgroups often excluded in RCTs. RWS are also useful for evaluating hard outcomes, long-term effectiveness, and safety [22]. Two types of RWS with SGLT-2is have been published: studies focusing on mortality, CV outcomes, and safety, generally conducted on large national or insurance company databases; and smaller classic cohort studies focusing on effectiveness and safety.

The first group of publications attempts to mimic the randomization of clinical trials through propensity score matching, which compares new SGLT-2i users with matched new users of other GLDs [23,24,25,26,27,28]. This kind of retrospective studies provides baseline patients’ characteristics and final hard outcomes, but drug effectiveness or medical interventions during the follow-up are normally not reported. These reports have confirmed, in a broader population of patients, the CV and renal benefits shown previously by SGLT-2is in RCTs.

The second group of RWS includes classic cohort studies, whose objective is to assess the effectiveness and safety of the SGLT-2is. These publications contain detailed information on patient characteristics, prescription orders, laboratory results, adverse effects (AEs), and withdrawals (WDs). We have identified several of these studies with canagliflozin, most of which are retrospective [29,30,31], except the Canadian Canagliflozin Registry, which was a national prospective study [32]. In all these reports, real-life effectiveness with both doses of canagliflozin (100 and 300 mg) was similar to those seen in RCTs.

Even though CANA300 seems to be more efficacious than canagliflozin 100 mg (CANA100) and other gliflozins, to date no RCTs or RWS have evaluated the strategy of intensification of SGLT-2i therapy by switching to CANA300, either from CANA100 or other SGLT-2is. The aims of this multicentric retrospective study, Real-World Evidence with Canagliflozin (Real-WECAN), were to assess in a real-world setting the effectiveness and safety of CANA100 daily as an add-on to the background antihyperglycemic therapy and to evaluate the intensification of SGLT-2i therapy by switching to CANA300 daily in patients with T2DM.

## 2. Material and Methods

### 2.1. Study Design and Patient Population

We conducted an observational, retrospective, multicenter study of two cohorts of adult patients with T2DM from 5 tertiary hospitals in the region of Madrid (Spain). A first cohort of SGLT2i-naïve patients who were initiated on CANA100 and a second cohort of patients with prior background SGLT-2i therapy who switched to CANA300 were identified in electronic medical records from the Departments of Endocrinology. The patients were consecutively selected from the diabetes clinic databases between May 2015 and July 2019 if they were older than 18 years, were diagnosed with T2DM according to the American Diabetes Association criteria, had received initial and regular follow-up diabetes care by endocrinologists in the diabetes clinic, started treatment with CANA100 or CANA300 as a part of their diabetes care at least 6 months before the data collection, and received at least 1 dose of the drug. Treatment decisions were in accordance with clinical practice and at the discretion of the treating physician. Switching to CANA300 was indicated in patients with suboptimal HbA1c or weight response to prior SGLT-2is. All patients were de-identified through assignment of unique patient study numbers to ensure confidentiality. Patients were followed until they discontinued therapy, died, or reached the end of the follow-up period. Discontinuation was defined as a period of at least 120 days without taking the medication. Participants who interrupted the drug at least once for ≥7 days but restarted it within the follow-up period were allocated to the interrupters group. Exclusion criteria were diagnosis of type 1 diabetes; missing data on HbA1c and weight at baseline or at the end of the follow-up; uncontrolled medical disease; and treatment with drugs that may induce impaired glycemic control (corticosteroids, immunosuppressants, somatostatin analogues, etc.).

### 2.2. Outcomes and Study Measures

Demographic, clinical, and laboratory data were extracted from electronic medical records by the investigators in each center, defining 3 data capture visits: V1, baseline (CANA100) or switch (CANA300); V2, 6 ± 2 months after the start of CANA100 or after switching to CANA300; and V3, last visit of the follow-up period. Electronic databases from primary care, laboratory departments, and emergency departments were also reviewed to collect unreported AEs.

Baseline clinical parameters included gender, age, duration of T2DM, microvascular and macrovascular complications, other CV risk factors (hypertension, hypercholesterolemia, hypertriglyceridemia, smoking), heart failure, arrhythmias, obstructive sleep apnea and background GLDs, and antihypertensive and lipid-lowering drugs. In the cohort of patients switching from other SGLT-2is to CANA300, clinical parameters before initiating the prior SGLT-2i were also registered. HbA1c, fasting plasma glucose (FPG), weight, BMI, systolic and diastolic BP (SBP and DBP), heart rate (HR), lipid profile, serum uric acid, hepatic serum biomarkers (alanine transaminase (ALT), aspartate aminotransferase (AST), gamma-glutamyl transferase (GGT), hematocrit, estimated glomerular filtrate rate (eGFR) using the Chronic Kidney Disease Epidemiology Collaboration equation, and microalbuminuria were collected at V1, V2, and V3. Echocardiographic data were recorded at baseline and over the follow-up in those cases where they were available. Causes of WDs and deaths were registered. AEs associated in clinical trials to SGLT-2is (genital mycotic infections (GMIs), urinary tract infections (UTIs), fractures, polycythemia, volume depletion, diabetic ketoacidosis (DKA), atraumatic lower extremity amputations) were collected by the investigators at each visit. GMIs and UTIs were diagnosed in those patients with positive cultures or patients reporting genital or urinary symptoms who received antifungal agents or antibiotics. Polycythemia was defined as increased hematocrit higher than 49% in men or higher than 48% in women. AEs related to volume depletion were diagnosed in patients with admissions in hospitals or emergency departments due to episodes of postural dizziness, hypotension, dehydration, or falls. DKA was defined as an anion-gap acidosis with increased plasma or urine ketones, even in the presence of normoglycemia. Hypoglycemia episodes were defined as biochemically documented (≤70 mg/dL) and severe hypoglycemia episodes, such as those requiring the assistance of another individual or resulting in seizure or loss of consciousness.

The main outcome measure was to assess changes in HbA1c at V2 and V3 from V1 in both cohorts of CANA100 and CANA300. Secondary outcomes were changes in weight, BP, lipid profile, serum uric acid, liver function tests, hematocrit, eGFR, and microalbuminuria.

### 2.3. Statistical Methods

Categorical data are shown in percentages. Continuous variables that follow a normal distribution are expressed as mean (standard deviation (SD)) and those that do not meet normality criteria are shown as median (interquartile range (IQR)). Analyses were carried out using the available data, without any imputation of missing data. Paired *t*-tests (for parametrically distributed data), Wilcoxon tests (for non-parametrically distributed data), and McNemar tests (for categorical variables) were performed to compare baseline data to those at follow-up. A subanalysis to compare those patients who switched from CANA100 to CANA300 with those individuals who remained on CANA100 over the follow-up was carried out using *t* tests for continuous variables and χ^2^ tests for categorical variables. We conducted bivariate comparisons of AEs and WDs between CANA100 and CANA300 cohorts using χ^2^ tests for categorical variables. A multiple linear regression analysis estimated the best predictive model of HbA1c reduction and WL at 6 months after switching from a prior SGLT-2i to CANA300. Control variables whose association with the response presented a *p* value > 0.1 were excluded from the multivariate analysis. The selection of the best regression equation for predictive purposes was performed using the Mallows criterion (lower Cp value), after previously constructing all possible submodels by combining the terms of the maximum model.

According to statistical data from previous observational studies [31,32], in order to detect statistically significant differences in HbA1c after 6 months of treatment with CANA100, assuming a minimum expected difference of −0.9%, SD of 2.5, 90% power, two-sided significance level of 0.05, and 20% premature WDs, 102 patients would be required. To detect statistically significant differences in HbA1c 6 months after switching to CANA300, assuming a minimum expected difference of −0.5%, SD of 2.5, 90% power, two-sided significance level of 0.05, and 20% premature WDs, 329 patients would be required. Overall, we needed to recruit at least 431 patients in the study.

All analyses were conducted with the Statistical Package for the Social Sciences (SPSS) version 15.0.1 (IBM Corp., Armonk, NY, USA) by using 2-sided tests and a significance level of 0.05.

The study was approved by the ethical review board (ethics code 19/56) of the centers which took part in the study and was done in compliance with the ethics guidelines for research in humans. All the procedures were in accordance with the requirements set out in the international standards for epidemiological studies, as recorded in the International Guidelines for Ethical Review of Epidemiological Studies and with the Helsinki Declaration of 1964, as revised in 2013. For this type of study, individual consent was not required.

## 3. Results

### 3.1. Demographic and Baseline Characteristics

In total, 583 patients met the inclusion criteria—279 in the CANA100 cohort (54.8% male, mean age = 59.7 years, mean HbA1c = 8.05%, mean BMI = 34.8 kg/m^2^) and 304 in the CANA300 cohort (55.9% male, mean age = 61.1 years, mean HbA1c = 7.51%, mean BMI = 34.5 kg/m^2^). In the second cohort, prior background SGLT−2i agents were: dapagliflozin 10 mg 51%, canagliflozin 100 mg 30.6%, empagliflozin 25 mg 11.1%, and empagliflozin 10 mg 7.3%. Median follow-up periods in both cohorts were 9.1 and 15.4 months, respectively. Baseline characteristics of the patients are shown in Table 1.

### 3.2. Analyses of Effectiveness

CANA100, as add-on to the background antihyperglycemic therapy, was associated with significant reductions in HbA1c at V2 and V3 (HbA1c −0.89% (95% CI: −1.06 to −0.72) and −0.90% (95% CI: −1.07 to −0.73), respectively, both *p* < 0.0001) (Figure 1). The percentage of patients with HbA1c below 7% significantly increased from 25.2% at V1 to 50.6% at both V2 and V3 (*p <* 0.0001). In the subgroup of patients with suboptimal glycemic control, defined as baseline HbA1c > 7% (*n* 197, mean HbA1c 8.68%), CANA100 decreased HbA1c by 1.17% at both V2 and V3 (*p <* 0.0001). In those patients with poor glycemic control, defined as baseline HbA1c > 8% (*n* 127, mean HbA1c 9.25%), CANA100 lowered HbA1c levels by 1.48% and 1.51% at V2 and V3, respectively (*p <* 0.0001). CANA100 was associated with significant WL at V2 and V3 (–3.8 kg (95% CI: −4.5 to −3.0)) and −4.1 kg ((95% CI: −4.9 to −3.3), respectively, both *p <* 0.0001) (Figure 1). There were also significant improvements in FPG, SBP, DBP, lipid profile, serum uric acid, and liver enzymes (Table 2). In those patients with baseline SBP > 140 mmHg (*n* 82, mean SBP 155.2 mmHg), CANA100 lowered SBP levels by −14.7 and −14.9 mmHg at V2 and V3, respectively (both *p* < 0.0001). A significant increase in hematocrit was observed over the follow-up. There were no changes in HR. CANA100 was associated with significant modest reductions in eGFR. No significant changes in albuminuria were observed in the entire cohort; however, in the subset of patients with micro- or macroalbuminuria, there was a significant decrease in albuminuria, from 74.9 mg/g at V1 to 38.7 and 26.9 mg/g at V2 and V3, respectively (both *p* < 0.01)

There was a slight increase in the use of metformin and GLP-1 receptor agonists (GLP-1ras) and a decrease in the use of sulfonylureas or glinides and dipeptidyl peptidase 4 inhibitors (DPP-4is) over the follow-up (Table 3). Daily insulin dose was reduced in patients treated with insulin at V1. A sensitivity analysis showed that in patients without GLP-1ra therapy, CANA100 was associated with a HbA1c reduction similar to that found in the entire cohort (−0.87%). However, the subgroup of patients who added GLP-1ras to CANA100 over the study showed a higher glycemic response (mean HbA1c reduction −1.61%). There were no meaningful changes in antihypertensive medications, including thiazides, loop diuretics, angiotensin-converting enzyme inhibitors (ACEis), and angiotensin II receptor blockers (ARBs). Prescription of lipid-lowering drugs slightly increased at V3.

In the CANA300 cohort, patients had previously been on SGLT-2i for a median period of 17.1 months before switching to CANA300. The baseline HbA1c and weight values of these patients when they initiated the former SGLT-2i were 7.92% and 95.7 kg, respectively (Figure 2). Significant reductions in HbA1c (−0.52%, *p* < 0.0001) and weight (−3.1 kg, *p <* 0.0001) were observed on prior SGLT-2i therapy before the switch (V1). After switching to CANA300, additional significant improvements in HbA1c and weight were found at V2 (HbA1c = −0.31% (95% CI: −0.42 to −0.20), *p <* 0.0001; weight −1.4 kg (95% CI: −1.9 to −0.9), *p <* 0.0001) and at V3 (–0.35% (95% CI: −0.47 to −0.23), *p <* 0.0001; weight −2.1 kg (95% CI: −2.8 to −1.5), *p <* 0.0001) (Figure 2). The percentages of patients with HbA1c < 7% significantly increased from 28.9% at V1 to 42.4% and 46.6% at V2 and V3, respectively (both *p <* 0.0001). In the subcohort of patients with suboptimal glycemic control, defined as HbA1c > 7% at V1 (*n* 205, mean HbA1c 8.41%), CANA300 decreased HbA1c by 0.46% at V2 and by 0.53% at V3 (both *p <* 0.0001). In those patients (*n* 72) with poor glycemic control, defined as HbA1c > 8% at V1 (*n* 72, mean HbA1c 8.94%), switching to CANA300 lowered A1C levels by 1.02% at V2 and to 1.12% at V3 (both *p <* 0.0001).

In the multivariate analysis, the only independent predictor of glycemic response at 6 months was high baseline HbA1c (–0.45% for each HbA1c percentage point). In the predictive model of weight reduction, only a diagnosis of sleep apnea (adjusted mean WL 1.4 kg higher than in patients without apnea) and background pioglitazone (adjusted mean WL 5.6 kg lower than in patients without pioglitazone) maintained the statistical significance in the model.

Overall changes in HbA1c and weight throughout sequential treatment with prior SGLT-2is and CANA300 (median follow-up 37.4 months (IQR 26.4–45.8)) were −0.83% (95%CI −0.94 to −0.60, *p <* 0.0001) and −5.1 kg (95% CI: −6.2 to −4.0, *p <* 0.0001), respectively. In the subgroup of patients who switched from CANA100 to CANA300, (median follow-up 31.7 months, IQR 20.6–40.1) additive changes in HbA1c and weight with both doses were −1.1% (CI 95% −1.4 to −0.70, *p <* 0.0001) and −5.5 kg (95% CI: −7.6 to −3.4, *p <* 0.0001), respectively. Patients who switched from CANA100 to CANA300 had a higher baseline HbA1c than those individuals who remained on CANA100 over the follow-up (8.50% vs. 7.81%, *p* < 0.0001), whereas other baseline characteristics and the glycemic and weight responses to CANA100 were not significantly different between subgroups.

There were also significant improvements in the FPG, SBP, DBP, lipid profile, and liver enzymes with CANA300 over the follow-up period (Table 2). In those patients with SBP > 140 mmHg at V1 (*n* = 71, mean SBP = 155.2 mmHg), CANA300 lowered SBP levels by −15.5 and –15.9 mmHg at V2 and V3, respectively (*p* < 0.0001). There were no significant changes in HR, hematocrit, or serum uric acid, although patients with uric acid >7 mg/dL at V1 experienced a significant reduction in urate levels from 8.3 to 7.3 mg/dL at V3 (*p* 0.012). Significant reductions in albuminuria, even in patients with normal UAE, were seen over the follow-up period. In the subset of patients with micro- or macroalbuminuria, UAE decreased from 82.2 mg/g at baseline to 39.8 and 40.2 mg/g at V2 and V3, respectively (both *p* 0.001). The eGFR did not significantly change at V2, although a slight decrease was observed at V3 (Table 2).

No meaningful changes in GLDs, antihypertensive medications, or lipid lowering drugs were observed after switching to CANA300 (Table 3).

### 3.3. Analyses of Safety

Drug discontinuation rates were 10.8% (CANA100) and 9.2% (CANA300) over the entire follow-up period (Table 4). In total, 33.6% patients treated with CANA100 switched to CANA300; 1.1% of patients on CANA100 and 1.0% of patients on CANA300 interrupted the drug at least once for ≥7 days but restarted it within the follow-up period. There were no deaths in the CANA100 cohort; 1 death (due to a stroke secondary to protein S deficiency) was reported in the CANA300 cohort, which was not considered by the investigator to be related to the study drug. The main reasons leading to CANA100 discontinuation (other than switching to CANA300) were GMIs (2.5%), patient decision (1.8%), bariatric surgery (1.4%), and UTIs (0.7%). The most frequent causes of WD with CANA300 were bariatric surgery (2.0%), UTIs (1.6%), GMIs (1.3%), and patient decision (0.7%).

The most common AEs with canagliflozin were GMIs (100 mg: 11.8%; 300 mg: 9.2%), hypoglycemia (100 mg: 8.2%; 300 mg: 9.2%; all of these cases were in patients with background insulin or SU), UTIs (100 mg: 4.7%; 300 mg: 7.2%), intravascular volume-related AEs (100 mg: 1.4%; 300 mg: 0.3%), fractures (100 mg: 1.1%; 300 mg: 0%), and polycythemia (100 mg: 0.4%; 300 mg: 1%). No DKA or amputations were reported in either cohort during the study. There were no significant differences in WDs or AEs between CANA 100 and CANA300 cohorts (Table 4).

## 4. Discussion

In this observational, retrospective, multicenter study, CANA100, as an add-on to the background antihyperglycemic therapy in SGLT-2i naïve patients with T2DM, and CANA300, as an intensification of prior SGLT-2i therapy, were associated with significant reductions in FPG, HbA1c, body weight, SBP, DBP, triglycerides, serum uric acid, and liver enzymes. Greater reductions in HbA1c and BP were seen among those patients receiving canagliflozin who had higher baseline HbA1c and SBP levels, respectively. In the subset of patients with albuminuria >30 mg/g at V1, there was a significant decrease in albuminuria with both doses, and switching to CANA300 lowered albuminuria even in patients with normal UAE.

In total, 33.7% of patients on CANA100 switched to CANA300; this subgroup had a higher baseline HbA1c than those patients who did not increase the canagliflozin dose, suggesting that physicians switched from CANA100 to CANA300 in those individuals who needed a greater HbA1c reduction to achieve their glycemic goals.

Our results in the CANA100 cohort are similar to those found in RCTs and other RWS. In placebo-controlled studies, CANA100 showed significant HbA1c reductions ranging from −0.63% to −0.89% at 18–26 weeks [33]. In several RWS, CANA100 lowered HbA1c by 0.72–0.98% after 6–12 months of follow-up [30,31,32,33,34,35]. In placebo-controlled studies, CANA100 decreased body weight by 1.8–3.7 kg and SBP was lowered by 3.3–5.3 mmHg [33]; similar results were found in RWS [31,32].

In our study, patients with prior background SGLT-2i therapy who switched to CANA300 showed modest improvements in HbA1c and weight. However, we should take into account that significant reductions in HbA1c and weight had already been attained with prior SGLT-2i therapy before the switch. In fact, HbA1c at V1 in the CANA300 cohort was lower than that observed in the CANA100 cohort and in RCTs. When we analyzed the subgroup of patients with poor glycemic control, CANA300 decreased HbA1c by 1.12% at V3. In the multivariate analysis, the only independent predictor of glycemic response after switching to CANA300 was high HbA1c at V1. This finding might be explained by the mechanism of action of SGLT-2is, as the degree to which UGE is increased in patients taking an SGLT2i is dependent in part on the degree of glycemia.

CANA300 appears to achieve greater HbA1c and weight reductions than other SGLT-2is according to indirect comparisons from phase-3 RCTs [21]. However, to the best of our knowledge, no RCTs or RWS had previously evaluated the strategy of switching to CANA300, either from CANA100 or other SGLT-2is in patients with T2DM. This approach could delay the need to intensify therapy, thus avoiding an increase in treatment burden. Switching between different classes of GLDs has successfully been tested in RCTs, replacing DPP-4is with GLP-1ras [36]. However, the concept of switching between GLDs from the same class is new and apparently counterintuitive. Notwithstanding, some RCTs have been published showing improvements in glycemic control after switching from short-acting GLP-1ras to long-acting GLP-1ras [37,38]; therefore, this approach could also make sense with SGLT-2is, because there seem to be some differences in effectiveness with CANA300.

In our study, both doses of canagliflozin improved serum liver enzymes. Pooled data from 6 RCTs showed that canagliflozin was associated with significant reductions in ALT, AST, and GGT in patients with T2DM compared with placebo or sitagliptin [39]. Changes in liver enzymes were fully attributed to changes in body weight and decreased HbA1c. In a recent prospective study, canagliflozin had beneficial effects on whole and segmental body composition, hepatic fat storage, and liver enzymes in patients with T2DM complicated by non-alcoholic fatty liver disease (NAFLD) for 12 months [40]. Similar results were found in a RCT with empagliflozin [41]. All these results suggest that SGLT2is promote fatty acid utilization and reduce subcutaneous, visceral, and hepatic fat in patients with T2DM and NAFLD.

CANA100 increased the hematocrit in our study; however, no additional effect was observed after switching from prior SGLT-2is to CANA300. In two exploratory analyses from the EMPA-REG OUTCOME trial and the CANVAS program, changes in hematocrit were the most important mediators of risk reduction of CV death with empagliflozin and heart failure with canagliflozin, respectively [42,43]. Enhanced delivery of oxygen to the tissues secondary to an increase in erythropoiesis has been postulated as a mechanism for the benefits of SGLT2 inhibition on CKD and CV disease.

Uric acid levels decreased in the CANA100 cohort and in those patients of the CANA300 cohort with high baseline uric acid values. Chronically elevated circulating uric acid concentrations are associated with increased risk of hypertension, CV disease, and CKD. Treating T2DM with an SGLT2i increases uric acid excretion, reduces circulating uric acid, and improves parameters of CV and renal function [44]. This raises the possibility that the lowering of uric acid by SGLT2is may assist in reducing adverse CV events and slowing the progression of CKD in T2DM. In the exploratory analysis from the CANVAS program, uric acid change was a significant mediator of the beneficial effect of canagliflozin on heart failure [43]. At present, however, there is no clear understanding of how uric acid lowering due to SGLT2 inhibition would drive reductions in heart failure or CV risk.

Significant reductions in albuminuria with both doses of canagliflozin were seen in the subgroup of patients with UAE > 30 mg/g at V1 over the follow-up period, in spite of the fact that there were no meaningful changes in antihypertensive medications, including diuretics, ACE, and ARBs, at V3. The natriuretic effects of SGLT-2is lead to increased delivery of sodium chloride at the macula densa, which reactivates tubular–glomerular feedback and reduces glomerular hyperfiltration and renal blood flow [2]. These alterations in renal hemodynamics with SGLT-2is present clinically as short-term reductions in albuminuria and eGFR, with subsequent stabilization of eGFR with long-term SGLT-2i therapy, and appear to be independent of improvements in blood glucose levels, weight, or BP.

Consistent with the safety findings observed in RCTs [33] and prospective RWS [32], the overall incidence of AEs with canagliflozin in our study was low, with most being mild or moderate in severity. We did not find a clear dose–response relationship in the frequency of AEs. The study discontinuation rate (excluding switching from CANA100 to CANA300) was approximately 10%, with the most frequent reasons associated with AEs being GMIs and UTIs. Interestingly, no DKA or amputations were reported in our study over the follow-up period. A significant increase in cases of DKA with SGLT-2i had been previously found in DECLARE and CREDENCE trials [13,14], although the number of patients was small and most of them were being treated initially with insulin. Canagliflozin was associated with increased amputation risk in the CANVAS program but not in CREDENCE, phase 3 clinical trials, or large RWS [26,45]; therefore, there is a possibility that the finding in CANVAS was the result of chance.

Our study has several limitations, many of which are due to its retrospective observational nature. We used a pre–post design to assess patient outcomes and did not include a control group. Selection bias is another potential limitation. All patients included in the analysis were treated in endocrinology clinics. On the one hand, these patients may have had more treatment-resistant T2DM than those in the primary care setting; on the other hand, they could have received a more comprehensive therapy for T2DM and obesity. Median follow-up time in the CANA100 cohort was shorter than in the CANA300 group (9.1 vs. 15.4 months), because 33.6% patients treated with CANA100 switched to CANA300. Since some GLDs were modified (either increased or decreased) over the follow-up period, especially in the CANA100 cohort, some influence of these changes on the final outcomes cannot be excluded, although several sensitivity analyses controlling for the effect of other GLDs did not modify the overall results. There is a potential recall bias in the frequency of AEs, as they were collected retrospectively. However, systematic research through electronic databases from primary care clinics, laboratory departments, and emergency departments was conducted in order to collect unreported AEs. Despite these limitations, well-designed descriptive observational studies can provide valuable information on real-world settings.

In summary, CANA100 (as an add-on therapy) and CANA300 (switching from CANA100 or other SGLT-2is) significantly improved several cardiometabolic parameters in patients with T2DM in a real-world setting, with a low incidence of AEs and a high rate of persistence. This study confirms the results from phase 3 RCTs with CANA100 and adds real-life evidence on the effectiveness and safety of switching to CANA300 from prior SGLT-2 therapy in patients with suboptimal metabolic control, which may avoid an increase in treatment burden.

## Figures and Tables

**Figure 1 jcm-09-02275-f001:**
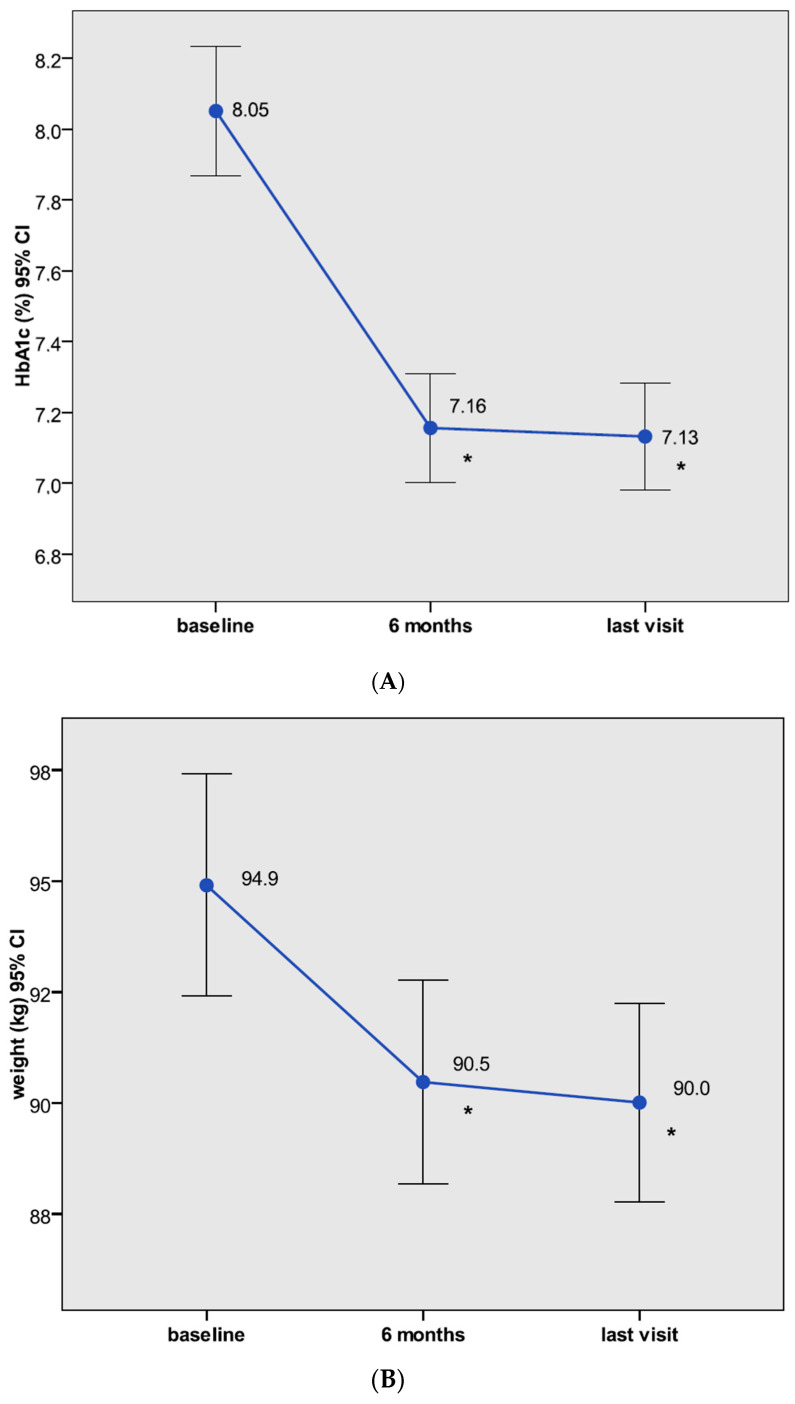
Changes in glycated hemoglobin (HbA1c) (**A**) and in body weight (**B**) after therapy with canagliflozin 100 mg at 6 months (V2) and at the end of the follow-up (V3). Data represent the mean (95% confidence interval); * *p* < 0.0001 vs. HbA1c and weight values at baseline (V1).

**Figure 2 jcm-09-02275-f002:**
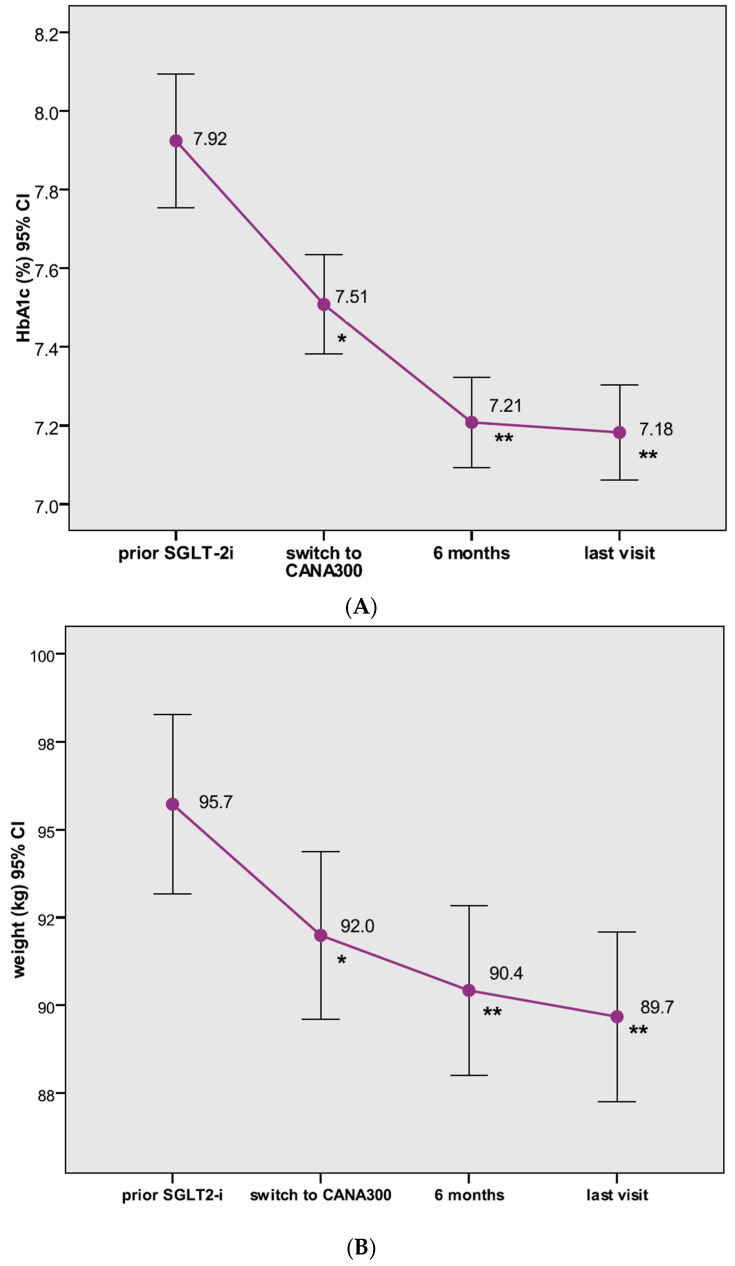
Changes in glycated hemoglobin (HbA1c) (**A**) and in body weight (**B**) from prior sodium–glucose co-transporter type 2 inhibitor (SGLT-2i) therapy to the switch to canagliflozin 300 mg (V1), and after switching to canagliflozin 300 mg at 6 months (V2) and at the end of the follow-up (V3). Data represent the mean (95% confidence interval); * *p <* 0.0001 vs. HbA1c and weight values at the initiation of prior SGLT-2i; ** *p* < 0.0001 vs. HbA1c and weight values at switch.

**Table 1 jcm-09-02275-t001:** Baseline characteristics of the cohorts of patients with canagliflozin 100 mg (CANA100) and canagliflozin 300 mg (CANA300). Data: percentage or mean (SD), except * median (IQR). ALT: alanine transaminase; ACEis: angiotensin-converting enzyme inhibitors; ARBs: angiotensin II receptor blockers; AST: aspartate aminotransferase; DBP: diastolic blood pressure; eGFR: estimated glomerular filtrate rate; GLDs: glucose-lowering drugs; GGT: gamma-glutamyl transferase; HDL-C: high-density lipoprotein cholesterol; LDL-C: low-density lipoprotein cholesterol; SBP: systolic blood pressure; V1: baseline (CANA100 cohort) or switch to canagliflozin 300 mg (CANA300 cohort); V2: 6 ± 2 months after the start of canagliflozin 100 mg (CANA100 cohort) or after switching to canagliflozin 300 mg (CANA300 cohort); V3: last visit of the follow-up period.

Baseline Characteristics	CANA 100 MG	Switch to Cana 300
Number of patients (*n*, %)		
Patients with data at V1	279 (100%)	304 (100%)
Patients with data at V2	265 (95.0%)	288 (94.7%)
Patients with data at V3	269 (96.4%)	294 (96.7%)
Follow-up time (months) *	9.1 (5.3–21.2)	15.4 (7.3–27.1)
Gender (male/female)	54.8/45.2	55.9/44.1
Age (years)	59.7 (12.4)	61.1 (10.3)
Duration of T2DM (years) *	10.4 (4.8–15.3)	12.5 (7.2–17.4)
HbA1c (%)	8.05 (1.53)	7.51 (1.12)
Patients with HbA1c > 7%	74.8%	71.1%
Fasting plasma glucose (mg/dL)	163.3 (57.5)	143.7 (35.9)
Weight (kg)	94.9 (21.1)	92.0 (21.1)
Height (cm)	165.3 (10.1)	163.6 (10.3)
BMI (kg/m^2^)	34.8 (7.2)	34.5 (7.5)
SBP (mmHg)	138.5 (20.2)	135.8 (14.5)
DBP (mmHg)	79.2 (10.1)	77.2 (11.0)
Heart rate (bpm)	84.6 (14.7)	84.9 (13.0)
LDL-C (mg/dL)	90.5 (39.5)	79.4 (24.8)
HDL-C (mg/dL)	42.0 (11.1)	45.0 (13.8)
Triglycerides (mg/dL) *	153.0 (114.0–227.5)	141.0 (108.0–195.8)
Uric acid (mg/dL)	5.5 (1.5)	5.1 (1.4)
Hematocrit (%)	44.3 (5.8)	45.5 (4.1)
AST (U/L) *	21 (16–27)	20.0 (15.5–25.5)
ALT (U/L) *	25 (18–36)	23.0 (17.0–30.0)
GGT (U/L) *	35 (22–58)	28.0 (19.0–44.0)
Serum creatinine (mg/dL)	0.84 (0.21)	0.85 (0.23)
eGFR (mL/min/1.73 m^2^)	86.5 (16.4)	84.9 (17.0)
Microalbuminuria (mg/g Cr) *	7.4 (4.1–31.1)	8.1 (0.0–26.3)
Diabetic renal disease (%)		
Stage G0/G1	51.7	47.6
Stage G2	37.3	43.2
Stage G3a	10.4	8.5
Stage G3b	1.9	0.3
Stage G4	0	0.3
Stage G5	0	0
Stage A1	74.4	79.3
Stage A2	19.9	18.0
Stage A3	5.8	2.7
Hypertension	73.5	83.6
Hypercholesterolemia	79.9	90.8
Hypertriglyceridemia	39.6	51.8
Combined hyperlipidemia	35.6	48.8
Current smoker	11.9	15.6
Ex-smoker	31.4	35.0
No smoker	56.7	49.3
Sleep apnea	17.3	24.3
Diabetic retinopathy	13.3	18.4
Diabetic renal disease	20.8	27.6
Diabetic neuropathy	7.9	11.5
Coronary artery disease	8.6	10.5
Stroke	3.9	3.6
Peripheral artery disease	3.2	9.2
Arrhythmias	4.7	4.6
Heart failure	1.8	3.0
Left ventricular hypertrophy (echo)	2.5	3.3
Glucose-lowering drugs		
Metformin	82.8	88.8
Sulphonylureas or glinides	18.6	9.9
Pioglitazone	1.4	3.0
DPP-4 inhibitors	41.6	24.3
GLP-1 receptor agonists	32.3	61.2
SGLT-2i	0	100
Dapagliflozin 10 mg	0	51.0
Empagliflozin 10 mg	0	7.3
Empagliflozin 25 mg	0	11.1
Canagliflozin 100 mg	0	30.6
Time with prior SGLT-2i (months) *	0	17.1
Insulin	38.4	42.1
Insulin therapy (years) *	5.3 (1.9–9.3)	6.4 (2.6–11.2)
Insulin dose(U/d)	56.5 (34.6)	46.2 (26.7)
Basal (%)	56.3	82.7
Basal-bolus (%)	43.7	17.3
Antihypertensive drugs		
0 (%)	31.5	21.4
1 (%)	22.9	28.9
≥2 (%)	45.6	49.7
ACEis (%)	32.9	31.0
ARBs (%)	30	44.6
Thiazides (%)	31.7	31.3
Loop diuretics (%)	5.8	6.6
Lipid-lowering drugs		
0 (%)	25.4	12.5
1 (%)	64.5	73.7
≥2 (%)	10.1	13.8

**Table 2 jcm-09-02275-t002:** Changes in blood pressure, heart rate, fasting plasma glucose, hematocrit, serum lipids, serum liver enzymes, renal function, and albuminuria in the cohorts of patients with canagliflozin 100 mg and canagliflozin 300 mg. Data: mean (standard error), except * median (IQR); ALT: alanine transaminase; AST: aspartate aminotransferase; DBP: diastolic blood pressure; eGFR: estimated glomerular filtrate rate; FPG: fasting plasma glucose; GGT: glutamyl transferase; HDL-C: high-density lipoprotein cholesterol; HR: heart rate. LDL-C: low-density lipoprotein cholesterol; NS: non-significant; SBP: systolic blood pressure; TGs: triglycerides; V1: baseline (canagliflozin 100 mg) or switch (canagliflozin 300 mg); V2: 6 ± 2 months after the start of canagliflozin 100 mg or switching to canagliflozin 300 mg; V3: last visit of the follow-up period.

	V1	V2	V3	*p* (vs. V1)
**CANAGLIFLOZIN 100 MG (*n* 279)**				
SBP (mmHg)	138.4 (1.5)	134.2 (1.0)	133.8 (1.3)	<0.0001
DBP (mmHg)	79.2 (0.7)	77.3 (0.7)	76.5 (0.7)	<0.01
HR (bpm)	84.6 (1.2)	83.3 (1.1)	83.2 (1.1)	NS
FPG (mg/dL)	163.2 (3.6)	132.1 (2.6)	132.7 (2.6)	<0.0001
Hematocrit (%)	44.3 (0.4)	46.1 (0.5)	46.1 (0.4)	<0.0001
LDL-C (mg/dL)	90.5 (2.5)	87.2 (1.9)	83.8 (1.8)	NS (V2) 0.013 (V3)
HDL-C (mg/dL)	42.0 (0.7)	43.1 (0.7)	44.6 (0.8)	NS (V2) < 0.0001 (V3)
TG (mg/dL)	189.1 (8.1)	179.6 (11.0)	177.1 (10.0)	<0.005
Uric acid (mg/dL)	5.5 (0.1)	5.0 (0.1)	5.0 (0.1)	<0.0001
AST (U/L)	25.8 (1.2)	22.5 (0.8)	23.0 (0.9)	<0.05
ALT (U/L)	30.6 (1.3)	26.9 (1.2)	26.3 (0.9)	<0.0001
GGT (U/L)	57.5 (5.3)	41.3 (3.3)	38.9 (3.0)	<0.0001
eGFR (mL/min)	86.3 (1.0)	84.1 (1.2)	84.4 (1.2)	<0.001
albuminuria (mg/g)	7.4 (4.1–31.1)	7.7 (3.0–22.2)	8.1 (2.0–20.6)	NS
**SWITCH TO CANAGLIFLOZIN 300 MG (*n* 304)**				
SBP (mmHg)	136.4 (0.9)	132.6 (1.0)	132.9 (1.1)	<0.005
DBP (mmHg)	77.0 (0.7)	75.4 (0.8)	73.8 (0.6)	NS (V2) < 0.0001 (V3)
HR (bpm)	84.9 (0.9)	83.8 (0.9)	83.9 (0.9)	NS
FPG (mg/dL)	144.1 (2.1)	134.8 (2.2)	132.7 (2.3)	<0.0001
Hematocrit (%)	45.5 (0.3)	45.5 (0.3)	45.9 (0.3)	NS
LDL-C (mg/dL)	79.4 (1.4)	78.1 (1.4)	75.7 (1.3)	NS (V2) 0.031 (V3)
HDL-C (mg/dL)	45.0 (0.8)	45.7 (0.9)	46.6 (0.8)	NS (V2) 0.047 (V3)
TG (mg/dL)	172.9 (8.7)	170.6 (7.1)	161.4 (6.6)	NS (V2) 0.033 (V3)
Uric acid (mg/dL)	5.1 (0.1)	5.1 (0.1)	5.1 (0.1)	NS
AST (U/L)	23.2 (1.3)	20.9 (1.0)	21.6 (1.0)	<0.01
ALT (U/L)	25.9 (0.8)	24.4 (0.8)	23.1 (0.7)	<0.005
GGT (U/L)	41.9 (2.9)	37.3 (3.1)	35.8 (2.4)	<0.05
eGFR (mL/min)	85.4 (1.0)	85.2 (1.0)	83.8 (1.0)	NS (V2) 0.006 (V3)
albuminuria (mg/g)	8.1 (0.0–26.3)	6.4 (0.0–18.1)	5.8 (0.0–15.4)	<0.001

**Table 3 jcm-09-02275-t003:** Changes in antihyperglycemic, antihypertensive, and lipid-lowering drugs in the cohorts of patients with canagliflozin 100 mg and canagliflozin 300 mg. Data represent a percentage or the mean (SD). ACEis: angiotensin-converting enzyme inhibitors; ARBs: angiotensin II receptor blockers; DPP-4is: DPP-4 inhibitors; GLDs: glucose-lowering drugs; GLP-1ras: Glucagon-like peptide-1 receptor agonists; V1: baseline (canagliflozin 100 mg) or switch (canagliflozin 300 mg); V2: 6 ± 2 months after the start of canagliflozin 100 mg or switching to canagliflozin 300 mg; V3: last visit of the follow-up period; * *p* < 0.0001 vs. V1; ** plus canagliflozin 100 mg from V2; *** *p* < 0.005 vs. V1.

Drug Class	V1	V2	V3
**CANAGLIFLOZIN 100 MG (*n* 279)**			
Metformin (%)	82.8	88.7	87.7
Sulphonylureas or glinides (%)	18.6	8.3	7.4
Pioglitazone (%)	1.4	1.9	1.5
DPP4is (%)	41.6	24.2	24.9
GLP-1ras (%)	32.3	47.5	49.4
Insulin (%)	38.4	37	36.3
Insulin dose (U/d)	56.5 (34.6)	48.7 (30.0) *	45.0 (30.4) *
Number of GLDs **	2.2 (0.9)	2.0 (0.9)	2.0 (0.9)
ACEis (%)	32.9	31.6	30.9
ARBs (%)	30.0	33.1	32.7
Thiazides (%)	31.7	28.3	28.0
Loop diuretics (%)	5.8	4.5	4.9
Number of antihypertensive drugs	1.4 (1.2)	1.4 (1.2)	1.4 (1.2)
Lipid-lowering drugs (%)	74.6	79.8	83.2
Number of lipid-lowering drugs	0.85 (0.6)	1.0 (0.6)	1.0 (0.6)
**SWITCH TO CANAGLIFLOZIN 300 MG (*n* 304)**			
Metformin (%)	88.8	89.7	88.8
Sulphonylureas or glinides (%)	9.9	7.9	8.8
Pioglitazone (%)	3	3.4	11.5
DPP4is (%)	24.3	19.9	20.7
GLP-1ras (%)	61.2	65.8	70.1
Insulin (%)	42.1	41.4	46.4
Insulin dose (U/d)	46.2 (26.7)	42.9 (27.1) ***	41.3 (29.0) ***
Number of GLDs	2.8 (0.9)	2.2 (0.9)	2.4 (0.8)
ACEis (%)	31.0	29.0	27.9
ARBs (%)	44.6	45.7	47.3
Thiazides (%)	31.3	28	29.2
Loop diuretics (%)	6.6	5.7	6.1
Number of antihypertensive drugs	1.6 (1.2)	1.6 (1.3)	1.6 (1.2)
Lipid-lowering drugs (%)	87.5	89.7	92.5
Number of lipid-lowering drugs	1.0 (0.5)	1.1 (0.5)	1.1 (0.5)

**Table 4 jcm-09-02275-t004:** Safety outcomes. * Withdrawals in the CANA100 cohort other than switching to CANA300. NA: not available.

	CANA100 (*n* 279)	CANA300 (*n* 304)	*p*-Value
Withdrawals (%) *	10.8	9.2	0.534
Deaths (%)	0	0.3	1.0
AEs of special interest			
Genital mycotic infections (%)	11.8	9.2	0.302
Hypoglycemia (%)	8.2	9.2	0.680
Urinary tract infections (%)	4.7	7.2	0.191
Intravascular volume-related AEs (%)	1.4	0.3	0.148
Fractures (%)	1.1	0	0.109
Polyglobulia (%)	0.4	1.0	0.916
Ketoacidosis (%)	0	0	NA
Amputations (%)	0	0	NA

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
