# Peer review of "Real-World Clinical Outcomes Associated with Canagliflozin in Patients with Type 2 Diabetes Mellitus in Spain: The Real-Wecan Study"

_jcm, 2020, doi:10.3390/jcm9072275_

Round 1

Reviewer 1 Report

This manuscript addresses in a real-world setting the effectiveness and safety of canagliflozin 100 mg/d (CANA100) as add-on to the background antihyperglycemic therapy, and to evaluate the intensification of prior SGLT-2 inhibitor (SGLT-2i) therapy by switching to canagliflozin 300 mg/d (CANA300) in patients with T2DM.

The multicentric retrospective study itself is carried out with careful treatments, but I think the approach is quite common, therefore not so much valuable information will be included in the draft.

Overall, I would suggest significant revision.

1.P1. (Title) The title should be shorter.

2.It would be clearer to give each abbreviation used with its explanation. 

3.P2. (Introduction) Several sentences that state the two types of RWS with SGLT-2is should be shortened.

4.P3. (Method) There is no explanation as to why they changed from CAN100 to CAN300. With the CAN100, did you switch to the CAN3000 because it wasn't effective enough? If so, the population characteristics of the CAN100 and CAN300 are different. This needs to be addressed as well.

5.p12 (Discussion) The discussion is redundant and difficult to understand. It should be expressed concisely.

6. To save space, combine Tables 1 and 2 in a table together.

Author Response

Reviewer 1 (please see more information in the attachment)

Point 1: (Title) The title should be shorter.

Response: The reviewer is right, and even though we would have liked to keep the concept of switch in the title, we must admit that it seems to be a bit complex to understand.

Changes:

  • We have shortened the title. REAL-WORLD CLINICAL OUTCOMES ASSOCIATED WITH CANAGLIFLOZIN IN PATIENTS WITH TYPE 2 DIABETES MELLITUS IN SPAIN: THE REAL-WECAN STUDY.

Point 2: It would be clearer to give each abbreviation used with its explanation

Response: We noticed that several abbreviations throughout the text did not have their explanations.

Changes:

  • We have added a list of all the abbreviations after the abstract in order to make the reading easier and included their explanations in the text (highlighted in red).

Point 3: (Introduction) Several sentences that state the two types of RWS with SGLT-2is should be shortened.

Response: We have to admit that the introduction was a bit long and, as the other reviewer pointed out, its structure should be improved.

Changes:

  • We have shortened several sentences regarding RWS with SGLT-2 (highlighted in red).
  • Besides, the introduction has been re-arranged, so the order of the paragraphs has been changed according to the recommendations of the reviewer 2:
  • Paragraph 1: SGLT-2is description and mechanism of action
  • Paragraph 2: SGLT-2is available in Europe and main pharmacological differences between CANA300 and the rest of SGLT-2is.
  • Paragraphs 3-4: RCTs and meta-analysis with SGLT-2i
  • Paragraphs 5-7: RWS.
  • As a result of this re-arrangement, some reference numbers have changed.

   We would welcome additional comments on this section if you have further concerns.

Point 4: P3. (Method) There is no explanation as to why they changed from CAN100 to CAN300. With the CAN100, did you switch to the CAN300 because it wasn't effective enough? If so, the population characteristics of the CAN100 and CAN300 are different. This needs to be addressed as well.

Response: Treatment decisions were in accordance with local clinical practice and at the discretion of the treating physician. It is common practice in the participating centers to increase canagliflozin dose in patients with suboptimal HbA1c and/or weight response. Notwithstanding, we have tried to clarify this point by conducting an additional statistical analysis comparing those patients on CANA100 who switched to CANA300 vs those ones who remained on CANA100 throughout the follow-up. The result is that the patients who switched from CANA100 to CANA300 had a higher baseline HbA1c than those individuals who remained on CANA100 (8.50% vs 7.81%, p<0.0001), whereas other baseline characteristics or the glycemic and weight response to CANA100 were not significantly different between both subgroups. Our interpretation of these findings is that physicians switched from CANA100 to CANA300 in those individuals with worse baseline control, so they needed a greater HbA1c reduction to achieve their glycemic goals.

Changes: All these findings have been added to the text.

  • MATERIAL AND METHODS. Study design and patient population. The following sentences have been included (highlighted in red):
    • Study design and patient population: “Treatment decisions were in accordance with clinical practice and at the discretion of the treating physician. Switching to CANA300 was indicated in patients with suboptimal HbA1c or weight response to prior SGLT-2is.”
    • Statistical methods: “A sub-analysis to compare those patients who switched from CANA100 to CANA300 with those individuals who remained on CANA100 over the follow-up was carried out using t tests for continuous variables and χ2 tests for categorical variables.”
  • Analyses of effectiveness. The following paragraph has been included (5th paragraph, highlighted in red). “Patients who switched from CANA100 to CANA300 had a higher baseline HbA1c than those individuals who remained on CANA100 over the follow-up (8.50% vs 7.81%, p<0.0001), whereas other baseline characteristics and the glycemic and weight response to CANA100 were not significantly different between both subgroups.”
  • The following paragraph has been included (2nd paragraph, highlighted in red). “33.7% of patients on CANA100 switched to CANA300; this subgroup had a higher baseline HbA1c than those patients who did not increase the canagliflozin dose, suggesting that physicians switched from CANA100 to CANA300 in those individuals who needed a greater HbA1c reduction to achieve their glycemic goals.”

Point 5: p12 (Discussion) The discussion is redundant and difficult to understand. It should be expressed concisely.

Response: We appreciate the reviewer comment. Several changes have been made to the discussion, such as deleting redundant numbers or results and re-arranging paragraphs to make some points easier to understand.

Changes:

  • The 3rd and 4th paragraph (in red) have been shortened and redundant results have been left out.
  • The paragraphs discussing lipids and multivariate analysis of weight loss have been deleted.
  • The paragraphs regarding hematocrit and uric acid (in red) have been re-written.
  • As a consequence, a reference regarding weight loss and pioglitazone has been deleted.

      We would welcome additional comments on this section if you have further concerns.

Point 6: To save space, combine Tables 1 and 2 in a table together.

Response: It makes sense to us, in fact the other tables actually combine both cohorts.

Changes:

  • Table 1 and Table 2 have been combined in the new Table 1 (highlighted in red)
  • Table 1 legend. Baseline characteristics of the cohorts of patients with canagliflozin 100 mg (CANA100) and canagliflozin 300 mg (CANA300).

Reviewer 2 Report

The authors performed a multicentric retrospective study to assess the effectiveness and safety in a real-world setting in two cohorts of T2DM, 1) canagliflozin 100 mg/d (CANA100) as add-on to the background antihyperglycemic therapy; 2) switching to canagliflozin 300 mg/d (CANA300) from prior SGLT-2 inhibitor (SGLT-2i) therapy. They concluded from their data that CANA100 (as add-on therapy) and CANA300 (switching from CANA100 or other SGLT-2is) significantly improved several cardiometabolic parameters in patients with T2DM in a real-world setting, with a low incidence of adverse effects and a high rate of persistence. Their study is agreed with the results from phase 3 RCTs with CANA100 and adds real-life evidence on the effectiveness and safety of switching to CANA300 from prior SGLT-2 therapy in patients with suboptimal metabolic control, which may avoid an increase in treatment burden. This study provided real-life evidence on the use of canagliflozin in term s of effectiveness and safety in T2DM patients. This is a very informative study and the reasons for performing such study is fairly straightforward. Here are several comments:

  1. Introduction is not organized logically. The logical arrangement can be, in the first 2 paragraphs to introduce the SGLT-2is and the use of SGLT-2is, while in the following paragraphs to introduce the RCTs and RWS for SGLT-2is.
  2. The figure 2, it seems that there are significant decreases of HbA1c and body weight at V1 compared to prior SGLT-2i, please calculate and, if any, add statistical analysis. Please also provide the details for CANA300 treatment at V1, i.e., the mean time for treatment of CANA300.
  3. It is very cleared for the safety analysis. It would be helpful to provide a figure or a table to show the statistical significance for the evaluation of safety in both cohorts.
  4. The discussion is too lone, try to shorten some paragraphs and arrange the paragraphs logically.
  5. Further check the language and terminology used in this manuscript.

Author Response

Reviewer 2  Please see more information in the attachment.

Point 1: (Introduction) Introduction is not organized logically. The logical arrangement can be, in the first 2 paragraphs to introduce the SGLT-2is and the use of SGLT-2is, while in the following paragraphs to introduce the RCTs and RWS for SGLT-2is.

Response: We would like to thank the comment. We have to admit that the introduction needs improving.

Changes: According to the other reviewer comments, we have shortened several sentences regarding RWS with SGLT-2. Besides, the introduction has been re-arranged, so the order of the paragraphs (highlighted in red) has been changed:

  • Paragraph 1: SGLT-2is description and mechanism of action.
  • Paragraph 2: SGLT-2is launched in Europe and main pharmacological differences between CANA300 and the rest of SGLT-2is.
  • Paragraphs 3-4: RCTs and meta-analysis with SGLT-2i.
  • Paragraphs 5-7: RWS.
  • As a result of this re-arrangement, some reference numbers have changed.

   We would welcome additional comments on this section if you have further concerns.

Point 2: The figure 2, it seems that there are significant decreases of HbA1c and body weight at V1 compared to prior SGLT-2i, please calculate and, if any, add statistical analysis. Please also provide the details for CANA300 treatment at V1, i.e., the mean time for treatment of CANA300.

Response: There was actually a significant difference between HbA1c and weight values at the initiation of the prior SGLT-2i and those ones at the time of switching to CANA300. In the results, the original manuscript already contained this sentence: “Significant reductions in HbA1c (-0.52%, p<0.0001) and weight (-3.1 kg, p <0.0001) were observed on prior SGLT-2i therapy before the switch (V1).”

Changes:

  • A new sentence depicting baseline characteristics of the CANA300 cohort at the initiation of the prior SGLT-2i has been included (highlighted in red). “In the CANA300 cohort, patients had been on the prior SGLT-2i for a median period of 17.1 months before switching to CANA300. Baseline HbA1c and weight of these patients when they initiated the former SGLT-2i were 7.92% and 95.7 kg respectively (Figure 2).”
  • Figure 2 has been modified including two different legends of statistical significance. *p<0.0001 vs HbA1c and weight values at the initiation of prior SGLT-2i. **p<0.0001 vs HbA1c and weight values at switch.

Point 3: It is very cleared for the safety analysis. It would be helpful to provide a figure or a table to show the statistical significance for the evaluation of safety in both cohorts.

Response: We have carried out a statistical analysis comparing side effects in both cohorts. There were no statistical differences between CANA100 and CANA300. In fact, we already mentioned this point (although without a formal statistical analysis) in the discussion. See the sentence “We did not find a clear dose-response relationship in the frequency of AEs.”

Changes:

  • The statistical analysis has been included in the Statistical methods section: “We conducted bivariate comparisons of AEs and WDs between CANA100 and CANA300 cohorts using χ2 tests for categorical variables.”
  • A new sentence has been added at the end of the Analyses of safety section: “There were no significant differences in WDs or AEs between CANA 100 and CANA300 cohorts (Table 4).”
  • A new table (Table 4) comparing withdrawals, deaths, and adverse effects of special interest has been included.
  • Legend to Table 4. “Safety outcomes. *Withdrawals in the CANA100 cohort other than switching to CANA300. NA: not available.”

Point 4: The discussion is too long, try to shorten some paragraphs and arrange the paragraphs logically

Response: We appreciate the reviewer comment (actually both reviewers agreed in this issue).

Changes:

Several changes have been made to the discussion, such as deleting redundant numbers or results and re-arranging paragraphs to make some points easier to understand.

  • The 3rd and 4th paragraph (in red) have been shortened and redundant results have been left out.
  • The paragraphs discussing lipids and multivariate analysis of weight loss have been deleted.
  • The paragraphs regarding hematocrit and uric acid (in red) have been re-written.
  • As a consequence, a reference regarding weight loss and pioglitazone has been deleted.

 We would welcome additional comments on this section if you have further concerns.

Point 5: Further check the language and terminology used in this manuscript

Response: A thorough revision of the manuscript searching for typos and misspellings has been made.

Changes:

  • We have added a list of all the abbreviations after the abstract in order to make the reading easier and included their explanations in the text (highlighted in red).
  • Several typos such as “diatolic”, “tubulo-glomerular” or “haemodynamics” have been corrected (in red in the text).

Round 2

Reviewer 1 Report

This second version of the paper is a great improvement, the authors are to be commended.